# Effective antibiotic dosing in the presence of resistant strains

**Asgher Ali**[1]*, **Mudassar Imran**[2], **Sultan Sial**[1], **Adnan Khan**[1]

**1** Department of Mathematics, Lahore University of Management Sciences, Lahore, Pakistan, **2** Department of Mathematics and Sciences, Ajman University, Ajman, UAE

* asgher.ali@lums.edu.pk

## Abstract

Mathematical models can be very useful in determining efficient and successful antibiotic dosing regimens. In this study, we consider the problem of determining optimal antibiotic dosing when bacteria resistant to antibiotics are present in addition to susceptible bacteria. We consider two different models of resistance acquisition, both involve the horizontal transfer (HGT) of resistant genes from a resistant to a susceptible strain. Modeling studies on HGT and study of optimal antibiotic dosing protocols in the literature, have been mostly focused on transfer of resistant genes via conjugation, with few studies on HGT via transformation. We propose a deterministic ODE based model of resistance acquisition via transformation, followed by a model that takes into account resistance acquisition through conjugation. Using a numerical optimization algorithm to determine the 'best' antibiotic dosing strategy. To illustrate our optimization method, we first consider optimal dosing when all the bacteria are susceptible to the antibiotic. We then consider the case where resistant strains are present. We note that constant periodic dosing may not always succeed in eradicating the bacteria while an optimal dosing protocol is successful. We determine the optimal dosing strategy in two different scenarios: one where the total bacterial population is to be minimized, and the next where we want to minimize the bacterial population at the end of the dosing period. We observe that the optimal strategy in the first case involves high initial dosing with dose tapering as time goes on, while in the second case, the optimal dosing strategy is to increase the dosing at the beginning of the dose cycles followed by a possible dose tapering. As a follow up study we intend to look at models where 'persistent' bacteria may be present in additional to resistant and susceptible strain and determine the optimal dosing protocols in this case.

## 1 Introduction

Misuse and overuse of antibiotics has led to bacteria developing resistance to antibiotics, and this has resulted in complications in treatment and even treatment failures. Antibiotic resistance is rising to very high levels throughout the world and is now a major public health challenge, with new resistance mechanisms emerging and spreading globally. According to the CDC, more than 2.8 million antibiotic-resistant infections occur every year in the US and

**Data Availability Statement:** All relevant data are within the manuscript.

**Funding:** The author(s) received no specific funding for this work.

**Competing interests:** The authors have declared that no competing interests exist.

account for more than 35,000 deaths. In the EU, antibiotic resistance causes more than 25,000 deaths every year, and there have been estimates attributing more than 700,000 deaths globally to antibiotic resistance [1–3].

Antibiotic resistance occurs when bacteria stop responding to antibiotic agents at specific concentration levels and are able to proliferate in the presence of the drug at these concentrations. While antibiotic resistance was reported very soon after the large scale use of antibiotics, this was ameliorated by the discovery of newer and more effective antibiotic agents; however, the development of new antibiotics has been outpaced by the acquisition of resistance to existing drugs by bacteria [4–6].

Bacteria can become resistant to antibiotics either by spontaneous mutations in their DNA (De-Novo resistance) or through the transfer of genetic material from drug-resistant bacteria, horizontal gene transfer (HGT). There are several mechanisms that give rise to resistance, and these include the production of enzymes that cause drug inactivation, alteration of the target sites of the drug, activation of efflux pumps that remove the drug from the bacterium, and the alteration of cell wall proteins preventing the uptake of the drug. The overuse and misuse of antibiotics provide selection pressure favoring the resistant strains causing these to proliferate. The well-known mechanisms for horizontal gene transfer include conjugation, transduction, and transformation. Conjugation or Bacterial conjugation involves the transfer of DNA via a plasmid during cell to cell contact. In the process of transduction, DNA is moved from one bacterium to another by a phage (virus), and in transformation, foreign genetic material is taken up and expressed by a bacterium. It has been well established in the literature [7] that HGT is the predominant mechanism for resistance acquisition; specifically, conjugation and transformation have been identified as primary pathways by which resistance may be transferred from a resistant strain to a susceptible one [8–12].

Several mechanistic models have been proposed in the literature to investigate the acquisition of antimicrobial resistance through HGT. These include models of HGT via conjugation, as well as via transformation. D'agata et al. presented an ODE based single and multidrug-resistant strain model with resistance acquired through conjugation which was modeled using a mass action law [13]. They included the innate immune response in their model, the effects of which were modeled using a Monod function. They studied that the the relationship between antimicrobial therapies immune response in preventing the proliferation of the resistant strains. Stekel et al. present a model of resistance via conjugation in a natural environment [14]. The model is based on a system of two ODEs and with conjugation modeled via a mass action term. They show that the spread of resistance is very sensitive to the rates of gene transfer and the antibiotic inflow. Imran et al. study the effects of different dosing regimens on a population of susceptible and resistant bacteria in a chemostat setting using a series of ODE based models [15]. Resistance may be acquired via bacterial conjugation which is modelled by a mass action term, effects of the antibiotic treatment are modeled by a pharmacodynamic function. The derive conditions for treatment success based on the model parameters. Svara et al. present a model to study selection pressure which favors the plasmid carried resistant genes in the presence of an antibiotic [16]. Using a system of ODEs to model the dynamics they show that the dosage and the inter-dose time of the antibiotic are critical in determining the selection for plasmid carried resistance. Torella et al. study the synergistic effects of multiple antibiotic agents and their efficacy, considering drug interaction [17]. They consider a susceptible bacterial strain, strains resistant to one particular antibiotic and strains that are resistant to both. The dynamics are modeled by a system of ODEs with the resistance to either strain acquired de novo. They include effects of drug interaction via a function representing the effective doses of the antibiotic felt by the susceptible strain. They show that greater synergy between the antibiotic may not always be beneficial and under certain conditions drug

antagonism may be more advantageous in preventing multi drug resistance. Johnson et al. look into dynamics that maintain natural competence (i.e. the ability to transform by uptake of DNA) and transformation in a bacterial population [17]. Their ODE based model incorporates compartments for bacteria that do and do not express competence and transformed bacteria, the process of transformation is modeled by a mass action law [18]. They show that selection pressure can help maintain competence in the population, this is further augmented if there is in addition selection pressure favoring transformants. Lu et al. explain the experimentally observed rates of natural transformation in the presence of motile and non-motile strains of Azotobacter vinelandii using a simple model of interaction between the competent cells and the transforming DNA based on Levin's mass action term [11].

The use of mathematical models to study antibiotic resistance and successful treatment protocols is an area of active research. The conventional treatment protocols involved periodic dosing with doses of constant strength, this, however, may not be the optimal strategy. Mathematical models have been used to identify successful strategies for antibiotic dosing. However, there does not seem to be a consensus on the issue. Several studies have recommended a hit-hard and fast strategy. This involves high early doses of the antibiotic [13, 19, 20], however other studies have shown that this may not always be optimal [21]. Some studies recommend a pulsed and tapered strategy, but other studies show that resistant strains may persist under such a regimen. Merideth et al. recommend dosing strategies be adopted in light of a pathogen-antibiotic interaction metric, based on the time a bacterial population takes to return to its initial density after a single dose of the antibiotic [21]. Bonhoeffer et al. investigated the use of single and multiple antibiotics. In the case of a single antibiotic, they find that high initial dosing is a better strategy [22]. Hoyle et al. study single and multiple fixed-dose regimens as well as tapered dosing [23]. They conclude that a single large dose is never optimal, while the optimal dosing when we also want to minimize the total antibiotic quantity follows a tapering pattern; in a follow-up work [24] they find similar results and also validate their findings using biological experiments. Penna-Miller et al. use optimal control to study dosing strategies in the case where commensal and pathogenic bacteria are present; they find that an 'intermittent' or pulsed dosing is optimal [25].

In this study, we consider the problem of determining the optimal antibiotic dosing when both susceptible and resistant bacteria are present. The goal is to find the dosing regimen that is successful in eradicating the bacterial population while keeping the total administered antibiotic at a minimum. Resistant strains are assumed to have a higher minimum inhibitory concentration (MIC). This is included in our models in the pharmacodynamic term as described in detail in the sections below. We now briefly describe the structure of the paper.

In section (2), we describe a numerical optimal control scheme, the direct gradient descent method (DGDM); we will use this to determine the optimal antibiotic dosing. We then consider a bacterial growth model in a chemostat setting proposed in [26] and use the DGDM to obtain a discrete optimal dosing schedule. We compare our results to those obtained using a quasi-optimal strategy in the literature.

In the next section (3), we propose a model for resistance acquisition via transformation. This is an HGT mechanism, where DNA fragments containing the resistant genes are taken up and incorporated in the chromosome of susceptible bacteria. We incorporate the uptake of antibiotic by the bacteria in our model. We first consider discrete doses of antibiotics applied periodically, using the killing rates of the susceptible and resistant strains as bifurcation parameters. We show that the system exhibits bistability, i.e., a periodic dosing regimen may not be successful. We then formulate the dosing problem as an optimal control problem, and we want to minimize the bacterial population while keeping the total antibiotic quantity low. Using the DGDM, we determine a discrete optimal dosing regimen. We show that the optimal

treatment is successful in eradicating the bacteria for a wide range of scenarios, including different antibiotic killing rates and different relative costs of the antibiotic dosing. Finally, we consider a slightly different functional in our optimal control setup; here, we are interested in minimizing the bacterial population at the end of our simulated time while keeping the total antibiotic quantity low. We note that the initial dosing schedule is somewhat different in this case, followed by tapered dosing.

In section (4), we determine the optimal dosing for the case where HGT occurs via conjugation; we use a model presented in [15]. We set up the optimal control problem and use the DGDM to determine the best antibiotic dosing strategy. We again consider two different functionals, and note that we obtain the qualitatively similar result as in the case where resistance acquisition is via transformation. We also compare our results to those obtained in [26], who determined a quasi-optimal strategy for this case, and note that while our method also suggests a tapered dosing strategy, the total antibiotic needed is much lower.

In the last section (5), we summarize the study and present the main findings of this work.

## 2 Optimal antibiotic dosing using the direct gradient descent method

We give a brief overview of our numerical optimization scheme, the direct gradient descent method (DGDM). We then apply it to a simple antibiotic dosing problem to benchmark the method. Using the method, we determine a discrete optimal strategy and compare the results with those obtained using a quasi-optimal strategy proposed in the literature [26]. In subsequent sections, we will use the DGDM to find the best dosing strategy when resistant bacteria are also present.

### 2.1 Direct gradient descent algorithm

We consider the problem of minimizing a cost functional, subject to differential equation constraints. Mathematically, the functional to be minimized is of the form

$$\min F(u) = \int_{t=a}^{t=b} f(u, s_1, s_2, ..., s_k) \, dt$$

subject to differential equation conditions

$$\frac{ds_i}{dt} = g_1(u, s_1, s_2, ..., s_k) \qquad i = 1, 2, \cdots, k$$

with initial conditions

$$s_i = s_i^0$$

We now give a description of our numerical optimization scheme, the DGDM. The gradient (functional derivative) $\nabla F(u)$ of a functional $F$ can be found from the Taylor series

$$F(u + h) = F(u) + <\nabla F(u), h> + O(h^2)$$

where $h$ is a permissible variation about $u$ and $<, >$ is the inner product in the space being used. Different inner products for the same set of vectors will result in different gradients with different numerical properties. The gradient $\nabla F(u)$ points in the direction of greatest increase of $F$ in the inner product space in which $\nabla F(u)$ was calculated.

Zizza [27] and Gigena [28] discussed the minimization of a functional subject to constraints by taking the exterior derivative of the functional and of the constraints. The wedge product of the exterior derivative gives a gradient in the independent variable that is set equal to zero and

then solved. In our case, we use automatic differentiation to find the gradient with respect to the control variable and construct all other state variables from the control, as described below. This avoids the Lagrange multiplier as well as the explicit construction of the gradient in the space of only the control.

We consider the state variables as functions of the control variable $u$. That allows us to consider the system of the control and state variables as defining a manifold with local coordinates $u$. The numerical problem is then done on a grid with equally space nodes on the interval $[a, b]$. The algorithm is given below.

For each optimization step;

1. values of the state variables $s_i$ are found numerically at each node

2. these values are used in the numerical analogue of the functional $F$

3. the gradient $\nabla F(u)$ is calculated the value of the functional is reduced by replacing $u$ by $u - \lambda \nabla F(u)$.

4. process continues until a convergence criterion is met.

## 2.2 A discrete optimal dosing strategy for susceptible bacteria

We consider a chemostat-based model of bacterial growth, introducing antibiotic dosing through a pharmacodynamic term; this was proposed by [15]. The model consists of three different compartments, representing the nutrient, antibiotic, and bacteria concentrations.

Let $S$ denote the nutrient concentration, $A$ the antibiotic concentration, $u$ the susceptible bacterial population with no drug resistance. The model equations are

$$
\begin{aligned}
\frac{dS}{dt} &= d_s(S^0 - S) - \frac{1}{\gamma}G(S)u \\
\frac{dA}{dt} &= d_A\{A_0(t) - A\} - f(A)u \\
\frac{du}{dt} &= \{G(S) - d_u - K(S, A)\}u
\end{aligned}
\tag{1}
$$

Here the first equation shows the change in nutrient concentration. The first term $d_S S^0$ represents the flux of nutrients into the chemostat, $-d_S S$ and $G(S)u$ are the washout and removal of nutrients by the bacteria, respectively. Similarly, the only way for antibiotics to deplete is through consumption by susceptible bacteria and via washout. The time-dependent function $A_0(t)$ shows the antibiotic dosing strategy that is being employed. The susceptible bacterial population is changing due to the nutrient-dependent growth term $G(S) = \frac{mS}{a+S}$, constant dilution with rate $d_u$ and killing of the susceptible bacteria due to the antibiotic, represented by the term $K(S, A) = \frac{kSA}{(a+S)(L+A)}$ which is assumed to be dependent on nutrient as well as and antibiotic concentration.

The problem we consider is designing a dosing strategy that minimizes the bacterial population while keeping the total cost of antibiotic dosing low at the same time. Mathematically, the optimization problem is to minimize the objective functional

$$
J[A_0(t)] = \int_{t_0}^{t_f} \left(\frac{1}{2}W_a A_0^2(t) + u(t)\right)dt
\tag{2}
$$

subject to ODE system constraints (1).

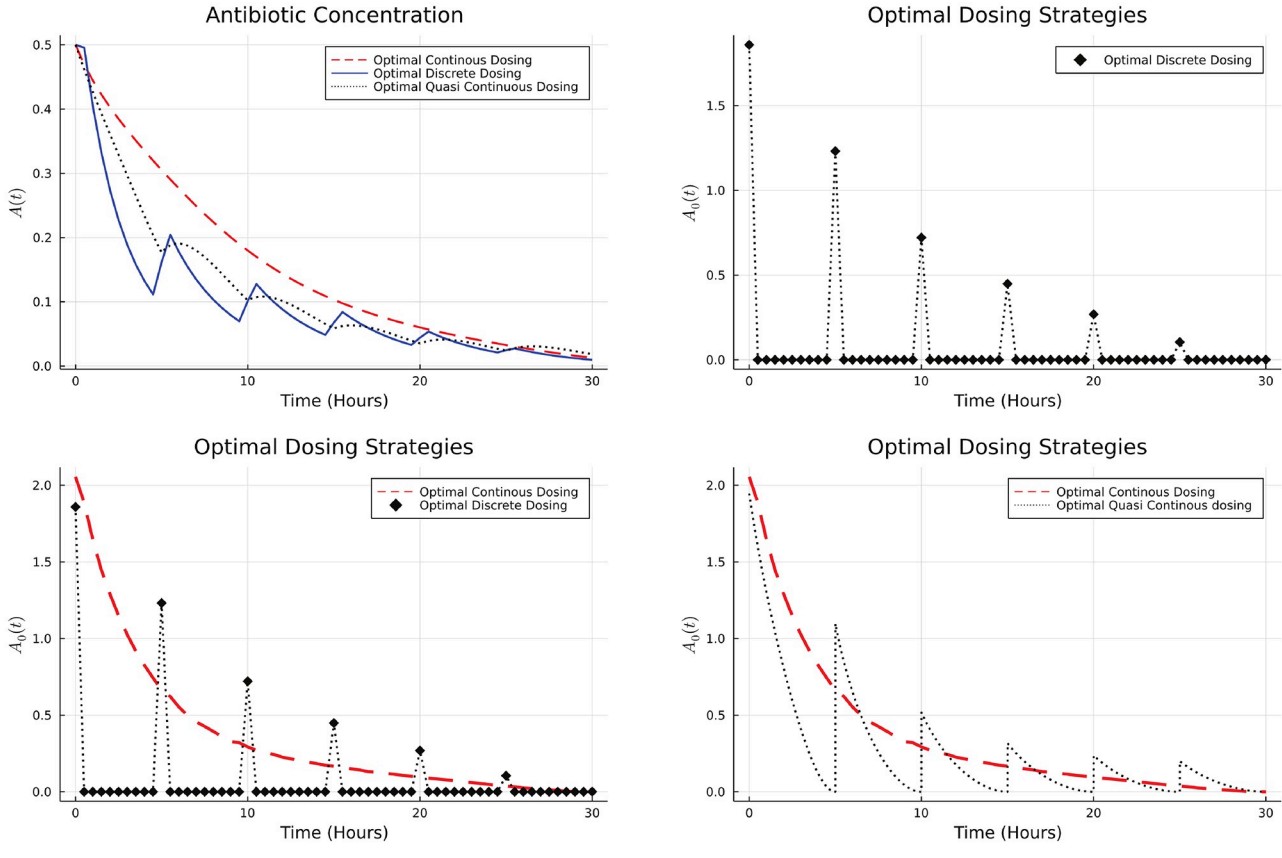

**Fig 1. Optimal dosing strategies and bacterial population.**

We use Pontryagin's Maximum Principle from the theory of optimal control to determine continuous and quasi-continuous optimal dosing strategies. In the case of quasi-continuous optimal control problem, we consider the $A_0(t) = A_m \delta(t)$ where $A_m$ is the maximum concentration of the antibiotic and $\delta(t)$ is a function which determines the antibiotic influx. For a detailed description of the numerical algorithm, we would refer to [26]. Next, we look at determining a discrete optimal dosing protocol using the DGDM. The underlying optimization problem is to minimize (2) subject to constraints (1). The dosing applied in this case is consists of discrete periodic doses given by $A_0(t) = \sum_i A_i \delta(t-iP)$. This means that the antibiotic concentration $A_i$ is applied periodically with a period $P$. For simulation purposes, we considered the case when $P = 5$ hours and the total number of cycles are 6. We compare the results obtained by continuous, quasi-continuous and discrete techniques in Fig 1.

The total antibiotic used at the optimal dosing is given in the Table 1.

We note that while all these methods lead to bacterial eradication, the discrete strategy does so at a much lower total antibiotic usage.

**Table 1. Total antibiotic consumption.**

| Numerical Scheme | $\int_{t_0}^{t_f} A_0(t)dt$ |
|---|---|
| Optimal Continuous Dosing | 10.5044 |
| Optimal Quasi Continuous Dosing | 7.3541 |
| Optimal Discrete Dosing | 1.8520 |

## 3 Modeling antibiotic resistance via transformation

Transformation is a horizontal gene transfer mechanism, where foreign DNA is taken up and incorporated by bacteria from the environment [12]. Within the body, cell-free DNA (cf-DNA) can be introduced into the blood due to cell death, active secretions, phagocytosis, autophagocytosis, etc. cf-DNA is distributed throughout the organism, and all cells release cf-DNA during cell division.

In this section, we propose a model of bacterial growth in a chemostat setting, where both resistant and susceptible bacteria are present; we also include the introduction of antibiotics in the system. The mechanism of resistance acquisition is horizontal gene transfer through transformation. Cell-free DNA can persist after disinfection and promote gene transfer in the absence of physical and temporal contact between a donor and recipient bacteria. Natural transformation is the process of physiological uptake of foreign DNA by 'competent bacteria,' and its genomic integration [29]. Competence is the ability of bacteria to take up extracellular DNA from the environment. Approximately 80 bacterial species, including human and animal pathogens, and soil bacteria, in their lifetime, are able to acquire the competence in a natural environment [30] and it has been posited that around 10 to 20% of a bacterial population may be competent [31].

We now define the state variables in the model. $S$ represents the nutrient concentration present in the chemostat, $A$ is the antibiotic concentration, $x$ the amount of cf-DNA present in the chemostat, and $u$, $v$ are the susceptible and resistant bacterial populations, respectively.

We hypothesize that the natural transformation happens through the uptake of cf-DNA from resistant strains by the competent bacterial population. In previous work on HGT via transformation, Johnson et al. [30] used the mass action dynamics to illustrate the transformation mechanism where the competent bacteria takes up the DNA interacting with the transformant bacterial population. Lu [11] proposed a mathematical model to explain the natural transformation dynamics in azotobacter vinelandii. The transformation mechanism used is based on Levin's mass action dynamics [18] and transformation rates for the tetracycline-resistant gene were estimated by taking into account the motile/non-motile nature of the bacterial population. Our work extends these models by including more of the process pathways in the model, in particular we incorporate a compartment for the cell free DNA density along with the resistant and susceptible bacterial populations which we hope better captures the dynamics, moreover we have set our model in a chemostat setting, as a proxy for a more complex organism, a detailed description of the model is given below

$$\frac{dS}{dt} = d_S(S^0 - S) - \gamma^{-1}\{G_1(S)u + G_2(S)v\}$$

$$\frac{dA}{dt} = d_A(A_0(t) - A) - f(A)\{u + v\}$$

$$\frac{dx}{dt} = K_2(S, A)v - d_x x - \alpha x u \tag{3}$$

$$\frac{du}{dt} = \{G_1(S) - d_u - K_1(S, A)\}u - k_3 x u$$

$$\frac{dv}{dt} = \{G_2(S) - d_+ - K_2(S, A)\}v + k_4 x u$$

where $G_1(S) = \frac{mS}{a+S}$, $G_2(S) = \frac{m_1 S}{a+S}$, $f(A) = \frac{vA}{L_1+A}$, $K_1(S, A) = \frac{kSA}{(a+S)(L+A)}$, and $K_2(S, A) = \frac{k_1 SA}{(a+S)(L+A)}$.

Here, $G_1(S)$ and $G_2(S)$ are the nutrient-dependent growth rates of the susceptible and resistant bacteria, and $K_1(S, A)$ and $K_2(S, A)$ are the antibiotic killing rates, which we have assumed to depend on the limiting nutrient and antibiotic levels, as has been done in the literature [32, 33].

The primary mechanism for nutrient concentration change includes a constant nutrient flux into the reservoir with a constant dilution of the nutrient and nutrient-dependent bacterial population growth.

The antibiotic concentration changes with the input of time dependent dosing given by $A_0(t) = \sum_i A_m \delta(t-iP)$ with a period $P$ over the time interval $[t_0, t_f]$. The antibiotic is consumed by the susceptible and resistant bacteria and its removal is represented by the term $-f(A)(u+ v)$.

The third additional state variable $x$ changes with the influx of cf-DNA from dead resistant bacteria. This depends on the killing rate $K_2(S, A)$ which results in the release of free circulating DNA. The term $-d_x x$ shows the constant dilution of the cf-DNA from the chemostat with dilution rate $d_x$. The term $\alpha x u$ represents the phenomena of transformation where the cell-free DNA is taken up and incorporated by the susceptible bacteria resulting in resistance acquisition.

The fourth equation shows the change in susceptible bacterial population density. This includes the nutrient-dependent influx, constant dilution with rate $d_S$, and nutrient and antibiotic-dependent killing of the susceptible bacterial population. The last term $-k_3 x u$ represents the removal of susceptible bacteria due to interaction with and incorporation of the cf-DNA leading to resistance.

The last equation represents the change in resistant bacterial density where $G_2(S)v$ is the growth of resistant bacteria at a nutrient-dependent rate. The terms $-d_+ v$, $-K_2(S, A)v$ show the removal of resistant bacteria due to washout and the effect of the antibiotic, respectively.

## 3.1 Periodic dosing treatment

We now consider antibiotic being given at fixed dosage periodically. We analyze the system and determine the steady states. We show that both sterile i.e. bacteria free steady state and infection states, those with the bacterial population surviving exist. We then find conditions for stability and the long term behavior of the system.

The parameters for the models are attached in Table 2.

**Theorem 3.1.** *The non-negative cone $\mathbb{R}_+^5$ is positively invariant for the model* (3) *and all solution for the model equations are ultimately uniformly bounded in forward time.*

*proof.* The proof is attached in the appendix A.1 in S1 Appendix.

The periodic solution to (3) includes the sterile steady state

$$E_0(t) = (S^0, A^*(t), 0, 0, 0) \tag{4}$$

where $A^*$ is the solution of $A'(t) = d_A(A_0-A(t))$. There are also other infection states where bacterial populations are not identically zero. The local stability of the sterile state can be found by the computing the Floquet exponents associated with the $u$ and $v$ states.

**Theorem 3.2.** *The sterile state $E_0(t)$ corresponding to* (3) *is locally asymptotically stable if $\lambda_4 = G_1(S^0)-d_u-[K_1(S^0, A^*)]_m < 0$ and $\lambda_5 = G_2(S^0)-d_+-[K_2(S^0, A^*)]_m < 0$ and unstable if either $\lambda_4 > 0$ or $\lambda_5 > 0$. Moreover in the constant case when $A_0(t) = A_0$ and both $\lambda_4 < 0$, $\lambda_5 < 0$ then $E_0$ is globally asymptotically stable.*

*Proof.* The proof is attached in the appendix A.2 in S1 Appendix.

To determine the stability of the sterile and infection states we use Floquet theory [34]. We calculate Floquet exponents and consider those related to the state variables $u$ and $v$, and if both of them are negative then the sterile state is stable and the system converges to the sterile

**Table 2. Description of the parameters used for models.**

| Parameters | Description | Values | |
|---|---|---|---|
| $S^0$ | Substrate feed concentration | 0.45 | [15, 26] |
| $A_m$ | Maximum antibiotic dosage concentration | 3 | [15, 26] |
| $d_S$ | Substrate dilution rate | 0.23 | [15, 26] |
| $d_A$ | Antibiotic dilution rate | 0.23 | [15, 26] |
| $d_u$ | Bacterial dilution rate | 0.23 | [15, 26] |
| $d_+$ | cf-DNA dilution rate | 0.23 | [15, 26] |
| $\gamma$ | Yield constant | 0.8 | [15, 26] |
| $q$ | Probability of Mis-segregation | 0.01 | [15, 26] |
| $\mu$ | Rate of plasmid transfer during Conjugation | 0.0000001 | [15, 26] |
| $\alpha$ | Re-Combination rate for Transformation | 0.0000001 | [11] |
| $m$ | Maximum growth rate for susceptible bacteria | 0.417 | [15, 26] |
| $m_1$ | Maximum growth rate for resistant bacteria | 0.416 | [15, 26] |
| $v$ | Maximum antibiotic uptake | 0.345 | [15, 26] |
| $k$ | Maximum kill rate for susceptible bacteria | 0.96 | [15, 26] |
| $k_1$ | Maximum kill rate for resistant bacteria | 0.87 | [15, 26] |
| $k_3$ | Susceptible bacterial removal rate via Transformation | 0.30 | Assumed |
| $k_4$ | Resistant strain formation rate | 0.15 | Assumed |
| $a$ | Half saturation constant for bacterial growth | 0.1 | [15, 26] |
| $L$ | Half saturation constant for bacteria kill | 0.1 | [15, 26] |
| $L_1$ | Half saturation constant for antibiotic uptake | 0.1 | [15, 26] |
| $W_a$ | Antibiotic cost sensitivity | 0.001 | [26] |
| $t_f$ | Final time (in hours) | 30.0 | [15, 26] |
| $P$ | Period of dosing regimens (in hours) | 5 | [26] |

state (Fig 2). If either one of these is positive the sterile state is unstable and the bacterial population grows (Fig 3). Our simulation results verify the theoretical findings.

Biologically this means that a periodic dose may or may not be successful in eradicating the bacteria, depending on the effect (killing rates) of the antibiotic on the susceptible and resistant bacteria and the amount of nutrient and antibiotic influx.

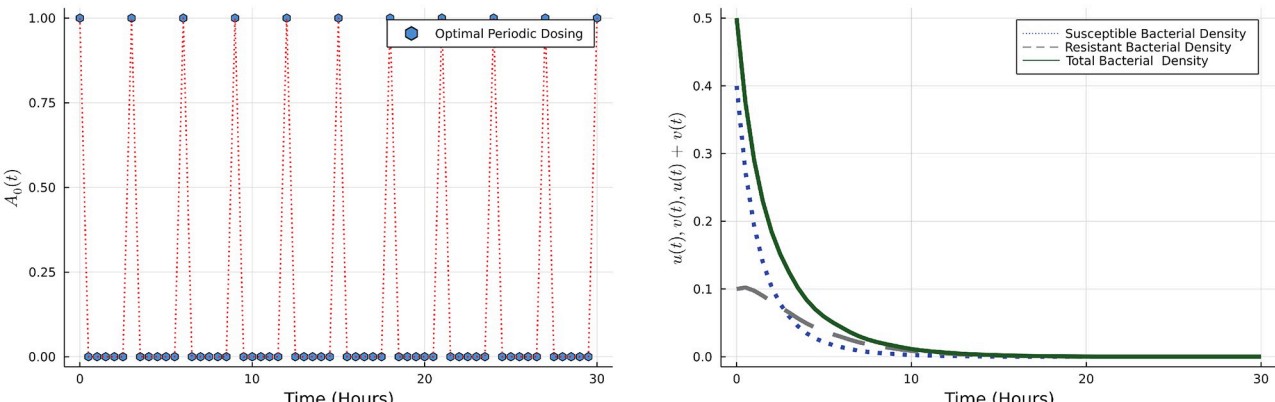

**Fig 2. The left figure shows the antibiotic usage throughout the time when $\lambda_4 = -0.1057, \lambda_5 = -0.0861$.** The right figure shows the success of the treatment for $0 < t < 30h$ when $\lambda_4 = -0.1057 < 0$ and $\lambda_5 = -0.0861 < 0$.

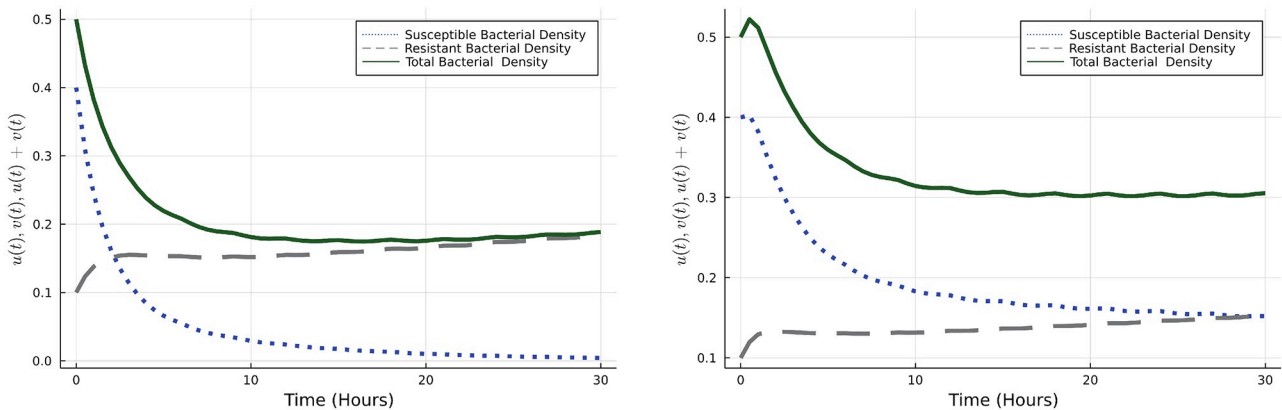

**Fig 3. The left figure shows the failure of the antibiotic treatment for $0 < t < 30h$ when $\lambda_4 = -0.0017 < 0$ and $\lambda_5 = 0.0821 > 0$.** The right figure shows the failure of the antibiotic treatment for $0 < t < 30h$ when $\lambda_4 = 0.3868$, $\lambda_5 = 0.4224 > 0$.

## 3.2 Optimal dosing strategy

**3.2.1 Minimizing the total bacterial population.** In this section, we look at the problem of determining the optimal dosing strategy, where the bacterial population in minimized while keeping the total antibiotic does low using the Direct Gradient Descent Method (DGDM). The functional we consider involves minimizing the total bacterial population along with the total antibiotic dosage. In the following section, we will look at the case where we consider minimizing the bacterial population at the end of the dosing period.

The mathematical problem is to find out the optimal antibiotic dosing strategy by minimizing the objective functional

$$J[A_0(t)] = \frac{1}{2} \int_{t_0}^{t_f} \left( W_a A_0^2(t) + u(t) + v(t) \right) dt \tag{5}$$

subject to constraints (3).

The dosing applied in this case is consists of discrete doses given by $A_0(t) = \Sigma_i A_i \, \delta(t - iP)$. This means that the antibiotic $A_i$ is applied periodically with a period $P$. For simulation purposes, we considered the case when $P = 5$ hours and total number of cycles are 6.

Using the DGDM, we determine the optimal dosing scheme and note that in this case a high initial dose with subsequent tapering is optimal (Fig 4). This is in line with several studies in the literature.

We now consider several scenarios and determine optimal dosing in each case. First, we vary the values of $k_+$, biologically as the value of $k_+$ becomes smaller the resistant strain is less

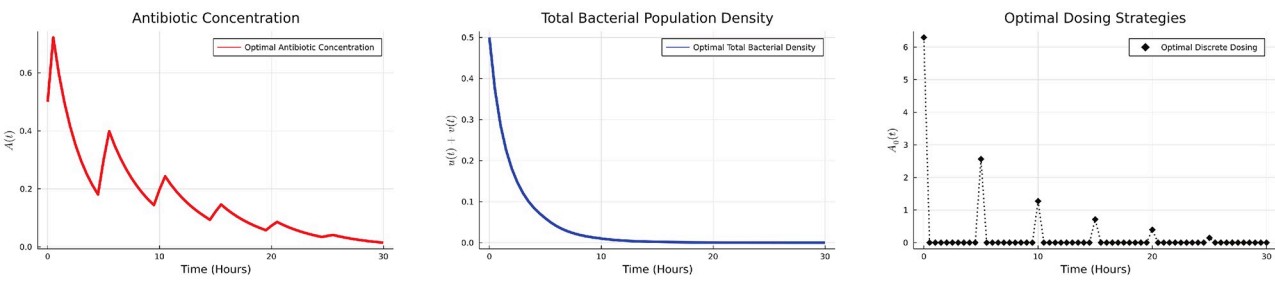

**Fig 4. Optimal dosing strategy and states.**

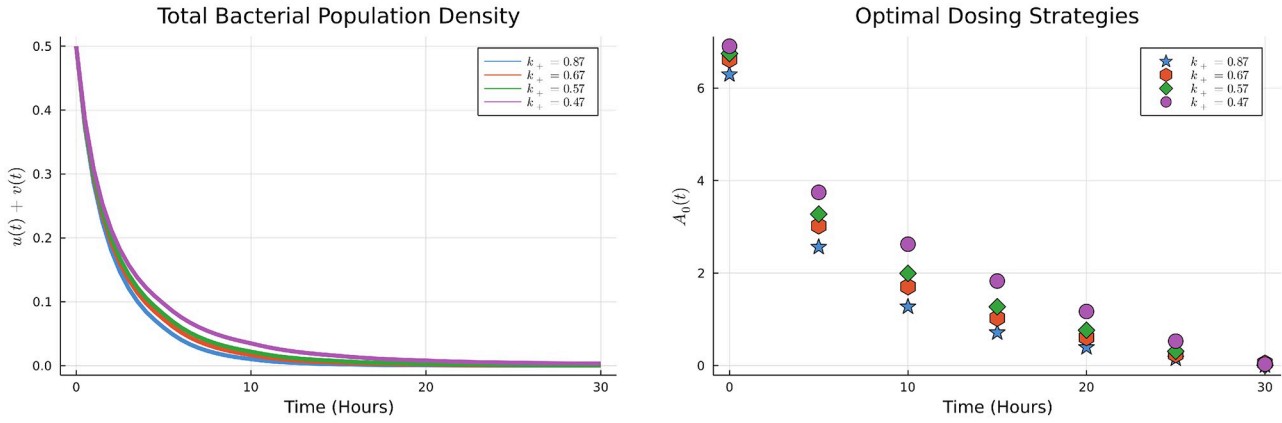

**Fig 5. Optimal bacterial populations and dosing strategies by varying the $k_+$.**

'susceptible' to the antibiotic. Using the DGDM, we obtain the dosing schemes under a range of values of $k_+$, the results are shown in the Fig 5.

Fig 5 shows that as the value of $k+$ is reduced, a higher total dose of antibiotic is needed. This is consistent over a wide range of values of $k_+$, moreover, a tapered dosing regimen is still optimal in each case.

Next, we look at how optimal dosing changes as the relative cost of the treatment is varied. In our model, this is captured by the parameter $W_a$ (Cost sensitivity parameter associated with the implementation of the dosing protocol). As expected, we observe that the total antibiotic in the optimal dosing is reduced as it becomes more costly. However, this also means that the bacterial population decreases at a lower rate as seen in the Fig 6. The dosing strategy remains qualitatively the same in this case, with higher initial doses that are tapered with time.

**3.2.2 Minimizing the bacterial population at the end of dosing period.** We now consider the problem of determining a dosing strategy where we want to minimize the bacterial population at the end of the dosing period, keeping the total antibiotic at a minimum. The mathematical problem is formulated by now evaluating $u$ and $v$ at the final time $t_f$ in the

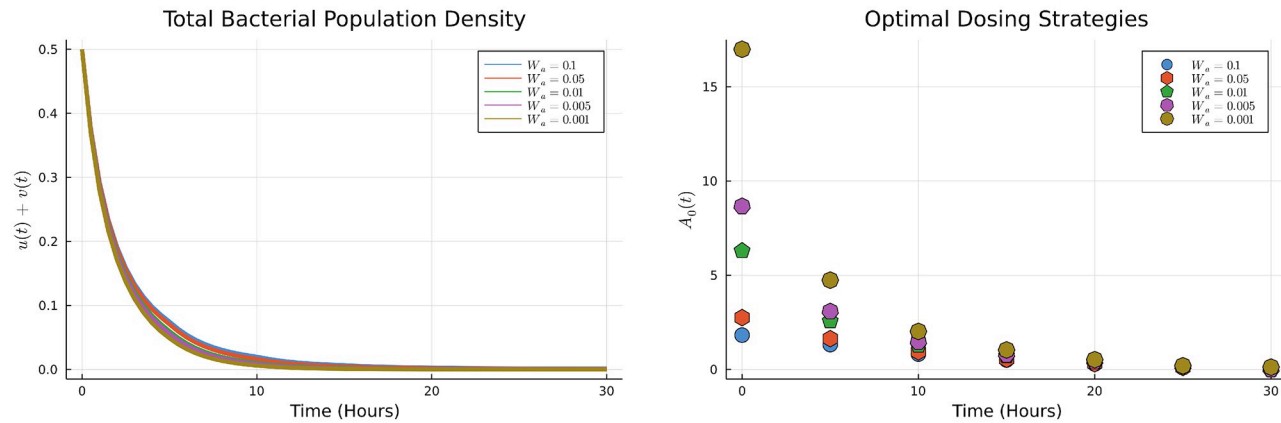

**Fig 6. Optimal dosing strategies and bacterial population corresponding to various cost sensitivity parameter $W_a$.**

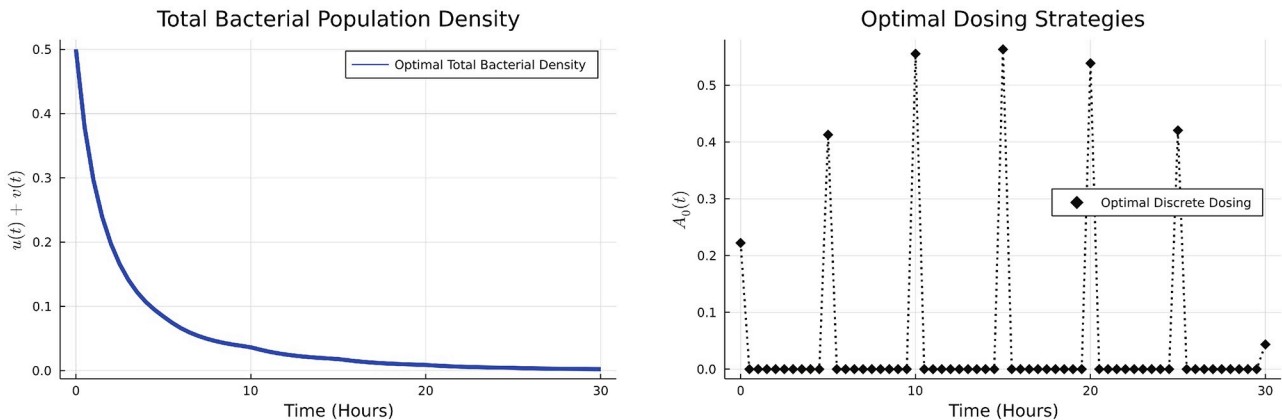

**Fig 7. Optimal dosing strategy and bacterial population.**

optimization functional as given below

$$J[A_0(t)] \quad = \quad u(t_f) + v(t_f) + \frac{1}{2}\int_{t_0}^{t_f} W_a A_0^2(t)dt \tag{6}$$

subject to ODE system constraints (3).

We use the DGDM to determine the optimal dosing strategy in this case, which is given in the Fig 7.

We now look into how the dosing schedule varies across different values of the antibiotic killing rate for resistant bacteria $k_+$, and for different relative antibiotic costs. Our results show that the optimal strategy remains qualitatively similar and involves an initial dose build up followed by tapering. This is observed across a wide range of values of $k_+$ and $d_u$ and demonstrated in Fig 8.

## 4 Modeling antibiotic resistance via conjugation

We consider a model for resistance acquisition through conjugation proposed by Imran et al. [15]. In follow-up work, Khan et al. used optimal control theory to determine a quasi-optimal

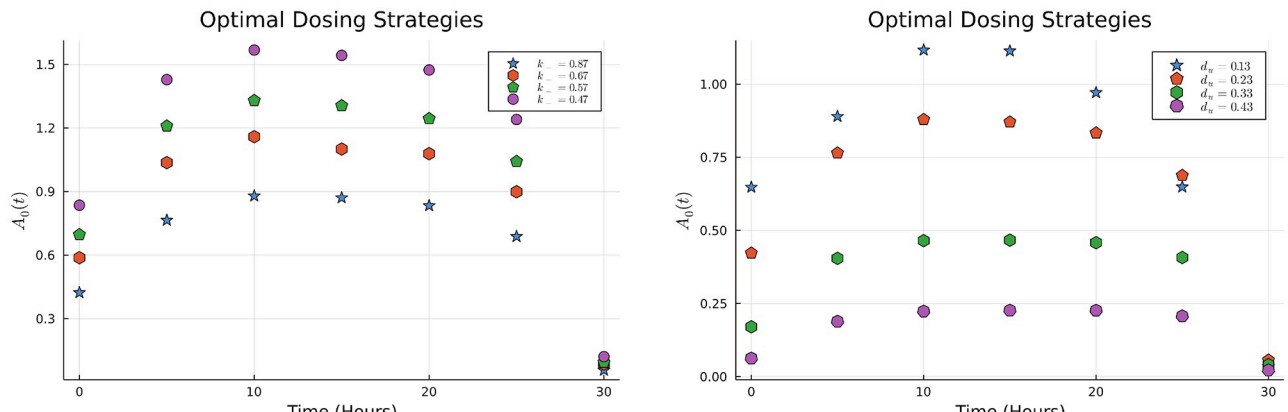

**Fig 8. The left figure shows the optimal dosing strategies for various $k_+$.** The right figure shows the dosing strategies for various values of washout parameter $d_u$.

dosing strategy for the problem [26]. Using the DGDM, we will find the optimal strategy and compare our results to those in [26]. We now give a brief description of the model.

Let $S$ denote the nutrient concentration, $A$ the antibiotic concentration, $u$ the susceptible bacterial population with no drug resistance, and $v$ is the resistant bacterial population. The model equations are

$$
\begin{aligned}
\frac{dS}{dt} &= d_S(S^0 - S) - \frac{1}{\gamma}\{G_1(S)u + G_2(S)v\} \\
\frac{dA}{dt} &= d_A\{A_0(t) - A\} - f(A)\{u + v\} \\
\frac{du}{dt} &= \{G_1(S) - d_u - K_1(S,A)\}u + qG_2(S)v - \mu uv \\
\frac{dv}{dt} &= \{G_2(S)(1-q) - d_+ - K_2(S,A)\}v + \mu uv
\end{aligned}
\tag{7}
$$

The model represents bacterial growth in a chemostat setting. The first and second equations represent the change in the nutrient and antibiotic concentrations. The change in the susceptible bacterial population density is represented by the third equation. Susceptible bacteria increase at a nutrient dependent growth rate $G_1(S)$, moreover it was posited that mis-segregation of resistant strains also give rise to susceptible bacteria. The last equation shows population density change in the resistant bacteria. The term $G_2(S)(1-q)v$ represents the increase the population, where the loss due to mis-segregation is accounted for, the process of conjugation is modelled by the mass action term $\mu uv$. For a more detailed discussion on the model we would refer to Imran et al. [15].

## 4.1 Optimal discrete dosing

**4.1.1 Minimizing the total bacterial population.**   We would like to determine the best dosing strategy which minimizes the overall cost of antibiotic and the susceptible and resistant bacterial population density.

The optimization problem takes the following mathematical form where we want to minimize the objective functional

$$
J[A_0(t)] = \int_{t_0}^{t_f}\left(\frac{1}{2}W_a A_0^2(t) + u(t) + v(t)\right)dt
\tag{8}
$$

subject to ODE system constraints (7). We want to determine the optimal discrete dosing given by $A_0(t) = \sum_i A_i \delta(t-iP)$. The antibiotic concentration $A_i$ applied periodically with a period $P$ will be determined using the DGDM. For simulation purposes, we considered the case when $P = 5$ hours and total number of cycles are 6. The optimal dosing schedule, antibiotic concentration and the bacterial population are given in the Fig 9.

We note that the optimal strategy is to give a high initial dose which is then tapered off with each subsequent dose. We also compare the discrete optimal dose to the quasi optimal regimen obtained by the method used in [26]. We observe that the discrete optimal strategy is successful in eradicating the bacteria at a lower total antibiotic dose. As antibiotic is mostly administered in discrete doses (or doses given over short intervals) the discrete regimen may also help quantitatively determine the optimal strategy.

**4.1.2 Minimizing the bacterial population at the end of dosing period.**   We also look at the problem which is to minimize the susceptible and resistant bacterial at the end of the antibiotic dosing along with cost associated with administering the antibiotic. The optimization

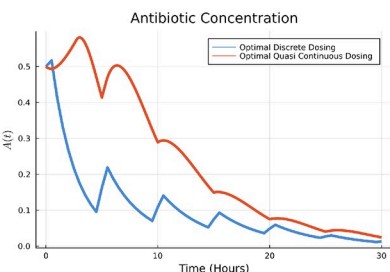
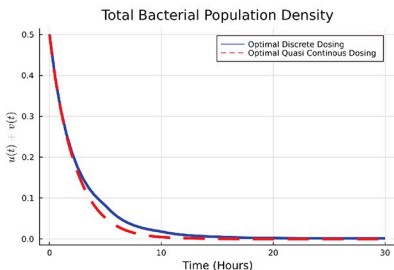
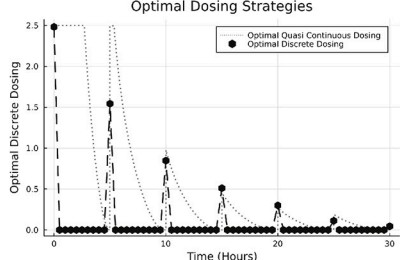

**Fig 9. Optimal dosing strategies, bacterial population, and antibiotic concentration.**

problem takes the following mathematical form

$$J[A_0(t)] \quad = \quad u(t_f) + v(t_f) + \frac{1}{2} \int_{t_0}^{t_f} W_a A_0^2(t) dt \tag{9}$$

subject to ODE system constraints (7). The optimal dosing strategies, antibiotic concentration and bacterial population are shown in Fig 10.

The optimal strategy in this case is an initial increase in the dose which is then tapered off. This is in contrast to the case where we want to minimize the total bacterial population. In that case, a higher initial antibiotic concentration is used but each subsequent does is smaller than the previous one. Qualitatively, both the results are similar for both HGT mechanisms we have studied.

## 5 Conclusion

In this work, we consider the problem of determining the optimal antibiotic dosing regimen when both susceptible and resistant bacterial strains are present. We consider mechanistic models for resistance acquisition and set up an optimal control problem, the goal being to minimize the bacterial population while keeping the total antibiotic dose low. We use a numerical optimal control method, the direct gradient descent (DGDM) algorithm, to determine these dosing regimens. Resistance acquisition is assumed to be via horizontal gene transfer. In particular, we propose a model of acquired resistance via transformation and determine the optimal dosing regimen for a variety of scenarios. We also consider a model from the literature for resistance acquisition via conjugation and find the best dosing strategies in this case.

We first describe our numerical optimal control algorithm, the DGDM, and then use it for the simple case of determining optimal dosing for susceptible bacteria. Using a chemostat-based model from the literature [15], we determine the dosing regimen and compare our results to some quasi-optimal ones in the literature [26].

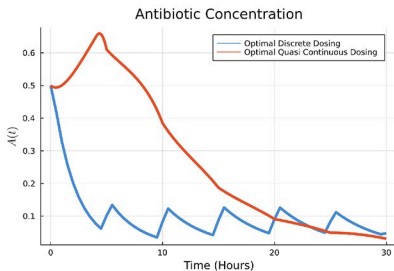
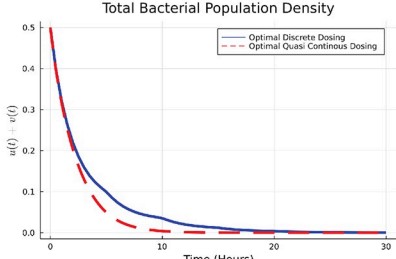
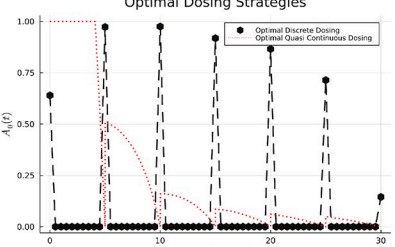

**Fig 10. Optimal dosing strategies, bacterial population, and antibiotic concentration.**

We then propose and analyze a model of resistance acquisition via transformation (3), where resistant genes are acquired by susceptible bacteria from the surrounding environment. We incorporate antibiotic dosing in the model and show that discrete antibiotic doses administered periodically may lead to treatment failure, i.e., the system exhibits bi-stability. This means that the success or failure of treatment under such dosing protocols would depend on the sensitivity of the resistant strain to the antibiotic.

We then set up the optimal control problem (3.2), where the dynamics of bacterial growth and antibiotic administration are governed by a mechanistic model. We want to determine a successful strategy to eliminate the bacteria while keeping treatment costs low by cutting down on antibiotic usage. We want to minimize the total bacterial population along with the administered antibiotic quantity. We look at different scenarios, varying the killing rates of the antibiotic for the resistant strain and varying the relative cost of administering the antibiotic. Our findings show that the optimal strategy in all cases is high initial loading followed by dose tapering.

We also look at the problem where we would like to minimize the bacterial population at the end of our treatment, rather than the total bacterial population over the dosing cycles. In this case, the optimal strategy is to build up the initial doses and then taper off. We again study the dosing under a variety of scenarios, varying the sensitivity of the resistant strains and the relative cost of antibiotic dosing. We note that qualitatively similar dosing is recommended in the different scenarios. A higher quantity of antibiotics is needed when the resistant strains are less sensitive. We also observe that as the relative cost of dosing becomes higher, the optimal dose is quantitatively smaller, the cost being that the bacterial population is reduced at a slower rate.

Finally, we consider treatment strategies when resistance is acquired through conjugation. We consider a model from the literature and set up the optimal control problem. Using the DGDM, we determine the optimal dosing once again for different scenarios. We find that the results are qualitatively similar to those in the transformation acquired resistance case.

We have determined discrete optimal antibiotic dosing when both resistant and susceptible bacteria are present. Our results can help determine both qualitative and quantitative dosing regimens for antibiotic treatment.

As a follow-up study, we would like to model horizontal gene transfer via transduction as a resistance acquisition mechanism. In this case, we would further determine efficient dosing protocols by setting up the optimal control problem and optimizing using our numerical optimization scheme. Another direction for future work is to extend our basic model and include a persister cell population, and this has been identified in the literature to be important in susceptible cell survival; the natural setting for this study would be a bio-film. We would again like to study efficient and effective antibiotic dosing based on the model.

## Supporting information

**S1 Appendix.**
(PDF)

## Author Contributions

**Conceptualization:** Asgher Ali, Mudassar Imran, Adnan Khan.

**Formal analysis:** Adnan Khan.

**Investigation:** Asgher Ali, Adnan Khan.

**Methodology:** Asgher Ali, Adnan Khan.

**Software:** Asgher Ali, Sultan Sial.

**Supervision:** Adnan Khan.

**Validation:** Sultan Sial.

**Writing – original draft:** Asgher Ali.

**Writing – review & editing:** Adnan Khan.

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
