## [Decision Letter · Decision Letter 0]

9 Aug 2022

PONE-D-22-16732Effective Antibiotic Dosing in the presence of Resistant StrainsPLOS ONE

Dear Dr. Ali,

Thank you for submitting your manuscript to PLOS ONE. After careful consideration, we feel that it has merit but does not fully meet PLOS ONE’s publication criteria as it currently stands. Therefore, we invite you to submit a revised version of the manuscript that addresses the points raised during the review process. Please submit your revised manuscript by Sep 23 2022 11:59PM. If you will need more time than this to complete your revisions, please reply to this message or contact the journal office at plosone@plos.org. Please include the following items when submitting your revised manuscript:A rebuttal letter that responds to each point raised by the academic editor and reviewer(s). You should upload this letter as a separate file labeled 'Response to Reviewers'.A marked-up copy of your manuscript that highlights changes made to the original version. You should upload this as a separate file labeled 'Revised Manuscript with Track Changes'.An unmarked version of your revised paper without tracked changes. You should upload this as a separate file labeled 'Manuscript'.

We look forward to receiving your revised manuscript.

Kind regards,

Ashwani Kumar, Ph.D.

Academic Editor

PLOS ONE

Journal Requirements:

Reviewers' comments:

Reviewer's Responses to Questions

**Comments to the Author**

1. Is the manuscript technically sound, and do the data support the conclusions?

Reviewer #1: Yes

Reviewer #2: Partly

2. Has the statistical analysis been performed appropriately and rigorously? 

Reviewer #1: N/A

Reviewer #2: Yes

3. Have the authors made all data underlying the findings in their manuscript fully available?

Reviewer #1: Yes

Reviewer #2: Yes

4. Is the manuscript presented in an intelligible fashion and written in standard English?

Reviewer #1: Yes

Reviewer #2: Yes

5. Review Comments to the Author

Reviewer #1: it is well written and well explained. The authors have clear aim and objective

If possible can the authors provide some pilot study data using the model DGDM. Please justify how this model will be better than already available systems.

Reviewer #2: Following recommendations are suggested to further enhance the manuscript

Kindly read the journal guidelines for articles submission and correct the formatting of your manuscript:

1) Kindly, rewrite the abstract, including its all contents, that introduction to literature review, study gaps, methodology, results, discussion, conclusion and future prospects, briefly in one paragraph.

2) Kindly arrange in-text references as per journal format.

3) Kindly give the line numbers as per journal format.

4) Kindly provide with lab-work pictures.

5) Arrange the references as per journal format.

6) Kindly include conclusion section.

7) add future prospects of this study.

8) Kindly mention the methodology and result section separately.

9) Kindly elaborate the literature review along with references.

10) Are the study gaps really come up with the need of this investigation. Kindly elaborate this point.

11) kindly include the models in literature used to carry out this study or those which are just mentioned in discussion section. It is recommended to briefly discuss each model from literature with your study models.

12) it is suggested to make a table in manuscript regarding the parameters of this study, instead of using them in appendix text.

6. PLOS authors have the option to publish the peer review history of their article (what does this mean?). If published, this will include your full peer review and any attached files.

Reviewer #1: No

Reviewer #2: No

---

## [Author Response · Author response to Decision Letter 0]

23 Aug 2022

We would like to thank the reviewers for the speedy turnaround and valuable comments. Their feedback has allowed us to improve the manuscript. We have updated the manuscript in light of the comments and a point by point response is given below. 

Editorial Comments: 

Response: The manuscript has been converted to the PLOS ONE format meeting the style requirements.

Response: We have updated the submission accordingly. 

3. In your Data Availability statement, you have not specified where the minimal data set underlying the results described in your manuscript can be found. 

Response: All data used in our simulation is available in the manuscript. 

Reviewer #1:

If possible can the authors provide some pilot study data using the model DGDM. Please justify how this model will be better than already available systems.

Response: In section 2 we provide a benchmark for the method by applying it to a simple well studied model. The two main contributions of the study are proposing a model for HGT via transformation and determining a discrete optimal dosing schedule. While there are many models in the literature that study HGT via conjugation, there are only a few models that look into HGT via transformation, further we have not seen any work on optimizing antibiotic dosing in this case. We also would like to note that in the case of our benchmarking model, the DGDM allows us to easily determine a discrete control (dosing regimen) for a continuous system, standard methods such as the Pontryagin’s Maximum Principle need significant modification to be applicable in such cases. 

Reviewer #2: 

1) Kindly, rewrite the abstract, including its all contents, that introduction to literature review, study gaps, methodology, results, discussion, conclusion and future prospects, briefly in one paragraph.

Response: The abstract has been updated in light of the comment.

2) Kindly arrange in-text references as per journal format.

Response: The manuscript has been updated in light of the comment.

3) Kindly give the line numbers as per journal format.

Response: The manuscript has been updated in light of the comment.

4) Kindly provide with lab-work pictures.

Response: Since this is a modeling and simulation study, the pictures are those obtained from simulation. We are planning a follow up work with a biologist to verify our model with in vitro experiments. 

5) Arrange the references as per journal format.

Response: The manuscript has been updated in light of the comment.

6) Kindly include conclusion section.

Response: The discussion section (5) was envisaged to be a conclusion section, the section title has been changed and it has also been updated. 

7) Add future prospects of this study.

Response: The conclusion section (6) has been updated with a discussion of future directions. 

8) Kindly mention the methodology and result section separately.

Response: Each of the sections where models are presented and discussed, are already organized in this manner. Since we present three different models, one of which has been formulated and studied in this work while we use the other two are used to present the optimal dosing strategy, it would be difficult to provide the methodology and results in two sections. As mentioned each of sections 3, 4 and 5 are already informally organized in this manner, where we present the methodology and then discuss the results. 

9) Kindly elaborate the literature review along with references.

Response: The introduction section has been updated, with more discussion of the models in the literature, in light of the comment.

10) Are the study gaps really come up with the need of this investigation. Kindly elaborate this point.

Response: This point has been addressed in the introduction, there are few modeling studies of antibiotic resistance acquired via transformation, and moreover to our knowledge no study has been done to look into optimal dosing treatments in this case. Our study is an effort in this direction. 

11) Kindly include the models in literature used to carry out this study or those which are just mentioned in discussion section. It is recommended to briefly discuss each model from literature with your study models.

Response: The introduction and conclusions sections have been updated, reference to the models mentioned in the conclusion have been added and more discussion of the models in the literature mentioned in the introduction has been added, in light of the comment.

12) it is suggested to make a table in manuscript regarding the parameters of this study, instead of using them in appendix text.

Response: The manuscript has been updated in light of the comment.

---

## [Decision Letter · Decision Letter 1]

2 Sep 2022

PONE-D-22-16732R1Effective Antibiotic Dosing in the presence of Resistant StrainsPLOS ONE

Dear Dr. Ali,

Thank you for submitting your manuscript to PLOS ONE. After careful consideration, we feel that it has merit but does not fully meet PLOS ONE’s publication criteria as it currently stands. Therefore, we invite you to submit a revised version of the manuscript that addresses the points raised during the review process. Please submit your revised manuscript by Oct 17 2022 11:59PM. If you will need more time than this to complete your revisions, please reply to this message or contact the journal office at plosone@plos.org. Please include the following items when submitting your revised manuscript:A rebuttal letter that responds to each point raised by the academic editor and reviewer(s). You should upload this letter as a separate file labeled 'Response to Reviewers'.A marked-up copy of your manuscript that highlights changes made to the original version. You should upload this as a separate file labeled 'Revised Manuscript with Track Changes'.An unmarked version of your revised paper without tracked changes. You should upload this as a separate file labeled 'Manuscript'.

We look forward to receiving your revised manuscript.

Kind regards,

Ashwani Kumar, Ph.D.

Academic Editor

PLOS ONE

Reviewers' comments:

Reviewer's Responses to Questions

**Comments to the Author**

1. If the authors have adequately addressed your comments raised in a previous round of review and you feel that this manuscript is now acceptable for publication, you may indicate that here to bypass the “Comments to the Author” section, enter your conflict of interest statement in the “Confidential to Editor” section, and submit your "Accept" recommendation.

Reviewer #1: All comments have been addressed

Reviewer #2: All comments have been addressed

2. Is the manuscript technically sound, and do the data support the conclusions?

Reviewer #1: Partly

Reviewer #2: Yes

3. Has the statistical analysis been performed appropriately and rigorously? 

Reviewer #1: N/A

Reviewer #2: Yes

4. Have the authors made all data underlying the findings in their manuscript fully available?

Reviewer #1: No

Reviewer #2: Yes

5. Is the manuscript presented in an intelligible fashion and written in standard English?

Reviewer #1: Yes

Reviewer #2: Yes

6. Review Comments to the Author

Reviewer #1: Authors have responded to the comment but have not given any satisfactory answer to the comment. Since there is almost no work on optimizing antibiotic dosing, wet lab experiments must be generated to veryfy the model. The validity of the model must be associated with realtime data. it seems to be a hypothetical model.

Reviewer #2: there are not further corrections required in manuscript. as all the recommended suggestions have been addressed.

7. PLOS authors have the option to publish the peer review history of their article (what does this mean?). If published, this will include your full peer review and any attached files.

Reviewer #1: No

Reviewer #2: No

---

## [Author Response · Author response to Decision Letter 1]

9 Sep 2022

We would like to thank the reviewers for the speedy turnaround and valuable comments. It seems like we may not have fully comprehended reviewer#1’s concern in our earlier response. A more detailed and hopefully pertinent response is given below. 

Reviewer #1: Authors have responded to the comment but have not given any satisfactory answer to the comment. Since there is almost no work on optimizing antibiotic dosing, wet lab experiments must be generated to verify the model. The validity of the model must be associated with realtime data. it seems to be a hypothetical model.

Response: Our model is built upon and extends previous work. Although there is not a great deal of literature on HGT via transformation, however, there are studies where models are presented. For example, [1] and [2] have proposed models for HGT via transformation. Our work extends the existing work by incorporating a compartment for cell-free DNA in the model, which we hope better captures the dynamics of the process. Moreover, we have set our model in a chemostat setting, as we want as a proxy for a more complex organism.

Further, we have presented some rigorous mathematical analyses of our model using Floquet theory. We show that the system is bi-stable under a constant periodic dosing regimen. This means that the same periodic dose may or may not be successful for different initial bacterial populations. This provides the rationale to look for optimal dosing regimens. 

We also would like to summarize some prior studies on optimal antibiotic dosing. Most of these are based on proposing a model for bacterial growth, including mechanisms for resistance acquisition and then searching (using different algorithms) for the optimal dosing, which results in bacterial eradication with low total antibiotic dosages. In [3], the authors propose an ODE-based model and its stochastic analog for Horizontal Gene Transfer (without specifying the exact mechanism) and then use genetic algorithms to search for the optimal dosing regimen. In [4], the authors study optimal antibiotic dosing strategies using an ODE-based model for bacterial growth in a chemostat setting and optimal control theory. In [5], a hybrid DE and Agent-based model is considered for TB, and the authors also use genetic algorithms to determine the optimal antibiotic dosing schedule. All these works are essentially modeling and simulation efforts (of course, the dynamics of bacterial growth and resistance acquisition (if considered) must be grounded in biology.

Moreover, the effects of the antibiotic need to be modeled by a proper pharmacodynamic function, which has been the case in all these studies as well as our work which is under consideration). They can provide a framework for wet lab experiments, as has been suggested by the reviewer. For example, in [6], the authors have actually used the framework presented in [3] for their wet lab experiments. 

We have also updated the section where we present our model in light of the comment. Further, the table of parameters has been updated as well. We hope that our response will be able to address the reviewer's concern.

References: 

[1] Lu N, Massoudieh A, Liang X, Kamai T, Zilles JL, Nguyen TH, et al. A kinetic model of gene transfer via natural transformation of Azotobacter vinelandii. Environmental Science: Water Research & Technology. 2015;1(3):363–374.

[2] Johnsen PJ, Dubnau D, Levin BR. Episodic selection and the maintenance of competence and natural transformation in Bacillus subtilis. Genetics. 2009;181(4):1521–1533

[3] Paterson IK, Hoyle A, Ochoa G, Baker-Austin C, Taylor NGH. Optimising Antibiotic Usage to Treat Bacterial Infections. Scientific Reports. 2016;6(1):37853

[4] Peña-Miller R, Lähnemann D, Schulenburg H, Ackermann M, Beardmore R. Selecting Against Antibiotic-Resistant Pathogens: Optimal Treatments in the Presence of Commensal Bacteria. Bulletin of Mathematical Biology. 2012;74(4):908–934

 [5] Cicchese, J.M., Pienaar, E., Kirschner, D.E. et al. Applying Optimization Algorithms to Tuberculosis Antibiotic Treatment Regimens. Cel. Mol. Bioeng. 10, 523–535 (2017)

[6] Hoyle A, Cairns D, Paterson I, McMillan S, Ochoa G, Desbois AP. Optimising eﬀicacy of antibiotics against systemic infection by varying dosage quantities and times. PLOS Computational Biology. 2020;16(8):1–20

---

## [Decision Letter · Decision Letter 2]

22 Sep 2022

Effective Antibiotic Dosing in the presence of Resistant Strains

PONE-D-22-16732R2

Dear Dr. Ali,

We’re pleased to inform you that your manuscript has been judged scientifically suitable for publication and will be formally accepted for publication once it meets all outstanding technical requirements.

Kind regards,

Ashwani Kumar, Ph.D.

Academic Editor

PLOS ONE

Additional Editor Comments (optional):

Reviewers' comments:

Reviewer's Responses to Questions

**Comments to the Author**

1. If the authors have adequately addressed your comments raised in a previous round of review and you feel that this manuscript is now acceptable for publication, you may indicate that here to bypass the “Comments to the Author” section, enter your conflict of interest statement in the “Confidential to Editor” section, and submit your "Accept" recommendation.

Reviewer #1: All comments have been addressed

Reviewer #2: All comments have been addressed

2. Is the manuscript technically sound, and do the data support the conclusions?

Reviewer #1: Yes

Reviewer #2: Yes

3. Has the statistical analysis been performed appropriately and rigorously? 

Reviewer #1: Yes

Reviewer #2: Yes

4. Have the authors made all data underlying the findings in their manuscript fully available?

Reviewer #1: Yes

Reviewer #2: Yes

5. Is the manuscript presented in an intelligible fashion and written in standard English?

Reviewer #1: Yes

Reviewer #2: Yes

6. Review Comments to the Author

Reviewer #1: The authors have clearly addressed the comments. it may be accepted for publication. congratulations to all the authors.

Reviewer #2: no further corrections needed. all queries were responded well. i would recommend to accept this manuscript.

7. PLOS authors have the option to publish the peer review history of their article (what does this mean?). If published, this will include your full peer review and any attached files.

Reviewer #1: No

Reviewer #2: No

---

## [Editor Report · Acceptance letter]

28 Sep 2022

PONE-D-22-16732R2 

Effective Antibiotic Dosing in the presence of Resistant Strains 

Dear Dr. Ali:

I'm pleased to inform you that your manuscript has been deemed suitable for publication in PLOS ONE. Congratulations! Your manuscript is now with our production department. 

Kind regards, 

on behalf of

Dr. Ashwani Kumar 

Academic Editor

PLOS ONE